# Direct correction of haemoglobin E β-thalassaemia using base editors

Mohsin Badat[1,2], Ayesha Ejaz[1], Peng Hua [1,10], Siobhan Rice[1], Weijiao Zhang[1], Lance D. Hentges [1,3,4], Christopher A. Fisher[1], Nicholas Denny [1], Ron Schwessinger[1], Nirmani Yasara [5], Noemi B. A. Roy[1], Fadi Issa [6], Andi Roy [1,7], Paul Telfer[2], Jim Hughes [1,3], Sachith Mettananda[5], Douglas R. Higgs [8] & James O. J. Davies [1,2,9] ✉

Haemoglobin E (HbE) β-thalassaemia causes approximately 50% of all severe thalassaemia worldwide; equating to around 30,000 births per year. HbE β-thalassaemia is due to a point mutation in codon 26 of the human *HBB* gene on one allele (GAG; glutamatic acid → AAG; lysine, E26K), and any mutation causing severe β-thalassaemia on the other. When inherited together in compound heterozygosity these mutations can cause a severe thalassaemic phenotype. However, if only one allele is mutated individuals are carriers for the respective mutation and have an asymptomatic phenotype (β-thalassaemia trait). Here we describe a base editing strategy which corrects the HbE mutation either to wildtype (WT) or a normal variant haemoglobin (E26G) known as Hb Aubenas and thereby recreates the asymptomatic trait phenotype. We have achieved editing efficiencies in excess of 90% in primary human CD34 + cells. We demonstrate editing of long-term repopulating haematopoietic stem cells (LT-HSCs) using serial xenotransplantation in NSG mice. We have profiled the off-target effects using a combination of circularization for in vitro reporting of cleavage effects by sequencing (CIRCLE-seq) and deep targeted capture and have developed machine-learning based methods to predict functional effects of candidate off-target mutations.

HbE has a particularly high prevalence in parts of the Indian sub-continent, China and Southeast Asia where up to 70% of the population are carriers due to the protection conferred against severe infection with malaria[1]. The HbE variant reduces production of β-globin chains and it may also form an unstable haemoglobin[2]. Co-inheritance of a severe β-thalassaemia mutation of the *HBB* gene on the other allele (HbE β-thalassaemia), can result in the need for blood transfusions every 2-3 weeks to sustain life[2]. Individuals with a severe β-thalassaemia mutation on one allele without the HbE mutation on the other have an asymptomatic carrier condition, known as β-thalassaemia trait. Gene therapy approaches have been developed for haemoglobinopathies, but these have significant safety concerns because an additional copy of the

[1]MRC Molecular Haematology Unit, MRC Weatherall Institute of Molecular Medicine, Radcliffe Department of Medicine, University of Oxford, Oxford, UK. [2]Department of Clinical Haematology, Royal London Hospital, Barts Health NHS Trust, London, UK. [3]MRC WIMM Centre for Computational Biology, MRC Weatherall Institute of Molecular Medicine, University of Oxford, Oxford, UK. [4]Oxford National Institute of Health Research Biomedical Research Centre, University of Oxford, Oxford, UK. [5]Department of Paediatrics, University of Kelaniya, Kelaniya, Sri Lanka. [6]Transplantation Research and Immunology Group, Nuffield Department of Surgical Sciences, University of Oxford, Oxford, UK. [7]Department of Paediatrics, University of Oxford, Oxford, UK. [8]Laboratory of Gene Regulation, MRC Weatherall Institute of Molecular Medicine, Radcliffe Department of Medicine, University of Oxford, Oxford, UK. [9]National Institute of Health Research Blood and Transplant Research Unit in Precision Cellular Therapeutics, Oxford, UK. [10]Present address: State Key Laboratory of Reproductive Medicine, Nanjing Medical University, Nanjing, China. ✉e-mail: james.davies@imm.ox.ac.uk

*HBB* gene is integrated randomly into the genome at thousands of different sites, carrying the potential for insertional mutagenesis and malignancy[3]. This is of particular concern as it would be deployed in children and haematopoiesis is a highly active process that requires a low mutational burden to develop malignancy compared to many other cell types[4]. Several methods have been described for genome editing for the treatment of β-thalassaemia. The approach that is closest to routine clinical implementation involves reactivation of fetal haemoglobin expression (HbF) either through mutagenesis of the promoters of the *HBG* genes or the erythroid enhancer of *BCL11A*[5–9]. Although these approaches are likely to lead to transfusion independence, they may not lead to an entirely normal phenotype due to the levels of HbF required to fully correct the pathophysiology of β-thalassaemia[10]. Furthermore, creation of a double-strand break carries risks due to deleterious on- and off-target repair outcomes and p53 inactivation[11]. We therefore aimed to develop a strategy for correcting the HbE mutation using direct editing of the affected codon with base editors. Here we show that it is possible to correct the HbE mutation directly using adenine base editors (ABEs) with high efficiency in patient derived CD34 + haemopoietic stem cells (HSCs) with minimal off target effects.

## Results

### Development of a base editing approach for HbE β-thalassaemia

The haemoglobin E codon (AAG) can be corrected to WT (GAG) or a variant haemoglobin, haemoglobin Aubenas (GGG) (Fig. 1a). The Hb

Aubenas variant has previously been reported to have a normal phenotype in a single family in heterozygosity although homozygous cases have not been reported (Supplementary Fig. 1a)[12]. HbE results in a mildly unstable haemoglobin and the mutation activates a cryptic splice site that causes abnormal mRNA processing[13]. The Aubenas variant is likely to be non-pathogenic because it introduces a glycine into an alpha helix on the external surface of the molecule and analysis with the machine learning model Splice AI[14] predicts that the cryptic splice site is removed (Supplementary Fig. 1b).

### Near complete editing of HbE in patient derived HSCs

Optimisation using different variants of base editors was undertaken initially in HUDEP-2 cells, WT CD34 + cells and patient CD34 + cells (Supplementary Fig. 2). Using ABE8-V106W we were able to achieve up to 98.8% correction (mean 90.2% SD 8.2%) of the HbE allele in CD34 + HSPCs from patients with HbE β-thalassaemia (Fig. 1b)[15]. The majority of edits converted the allele to Hb Aubenas (mean 78.0%) or to the WT sequence (mean 12%) (Fig. 1c). A potential editing outcome is an AGG codon, which has never been described in patient studies. Edited alleles including this codon were observed but at extremely low levels (mean 0.74% SD 0.64%) and are thus unlikely to be clinically significant.

Patients with the common IVS 1-5 β-thalassaemia mutation showed minimal editing (0.67%) of the thalassaemic allele due to disruption of the protospacer adjacent motif (PAM) by this variant. In

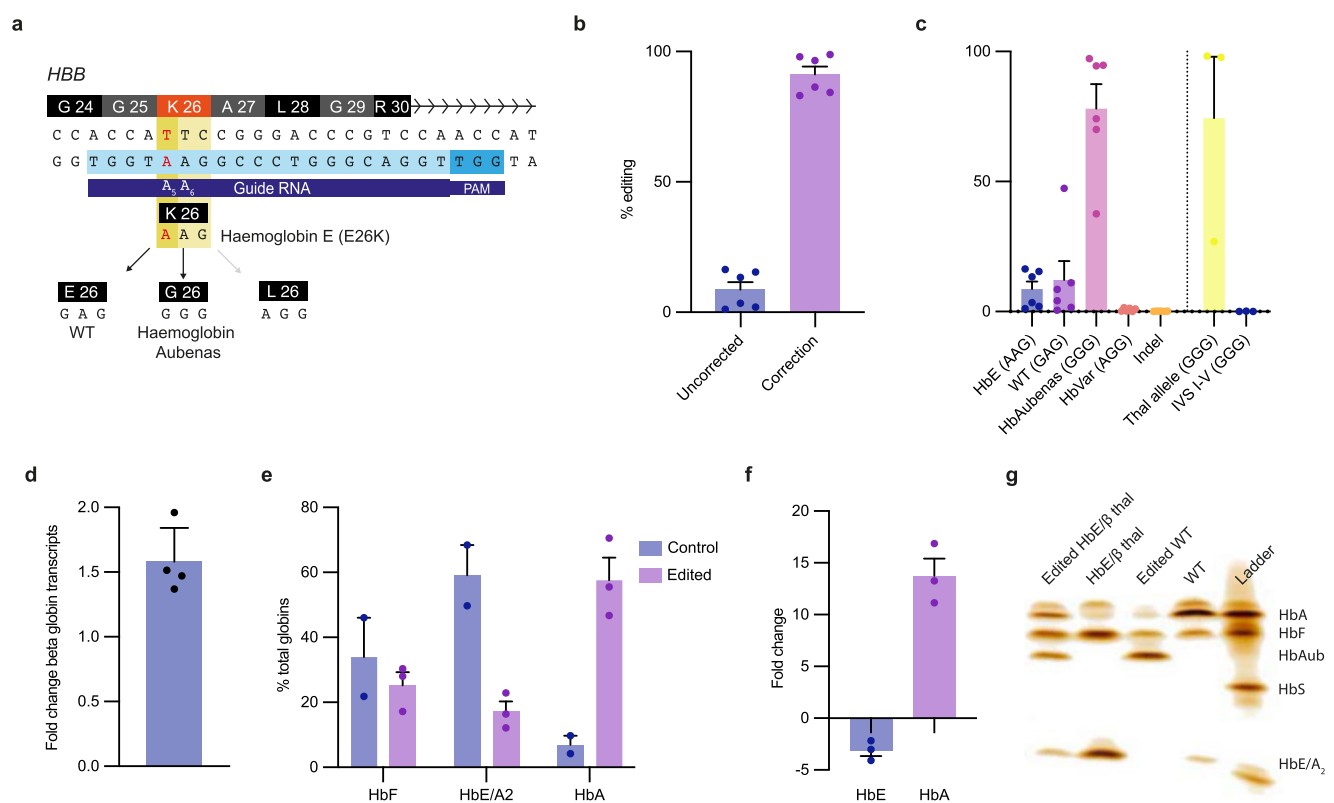

**Fig. 1 | Base editing of the Haemoglobin E. a** ABE editing strategy to repair the HbE mutation. The target adenine in HbE ($A_5$) lies at position 5 of the protospacer, with a bystander adenine ($A_6$) at position 6. The HbE codon (AAG) can be edited with one of three outcomes depending on which adenine has been deaminated: editing $A_5$ alone reverts the codon to WT (GAG), both $A_5$ and $A_6$ converts it to a normal variant that codes for β^Aubenas (GGG) and $A_6$ alone converts it to a previously undescribed codon (AGG). **b** Adenine base editing using ABE8e V106W highly efficiently converts the HbE codon to normal or a normal variant (*n* = 6 biologically independent samples). **c** Codon editing outcomes on the HbE allele and the non-target thalassaemic allele (where the thalassaemic allele was not sequenced editing at HBD

was used as a surrogate) (*n* = 6 biologically independent samples). **d** Increase in β-globin expression as assessed by the β/α ratio in edited erythroid cells from patients with HbE β-thalassaemia compared to unedited controls (*n* = 4 biologically independent samples). **e, f** Haemoglobin variants in control and edited erythroid cells measured by CE-HPLC. HbE and HbA₂, and HbA and Hb Aubenas are given together as they run in the same window and cannot be resolved using CE-HPLC (*n* = 3 biologically independent samples; *n* = 2 for unedited controls). **g** IEF showing haemoglobin variants in control and edited cells. Hb Aubenas is clearly detected but no other novel haemoglobins are observed. All error bars represent the standard error of the mean. Source data are provided as a Source Data file.

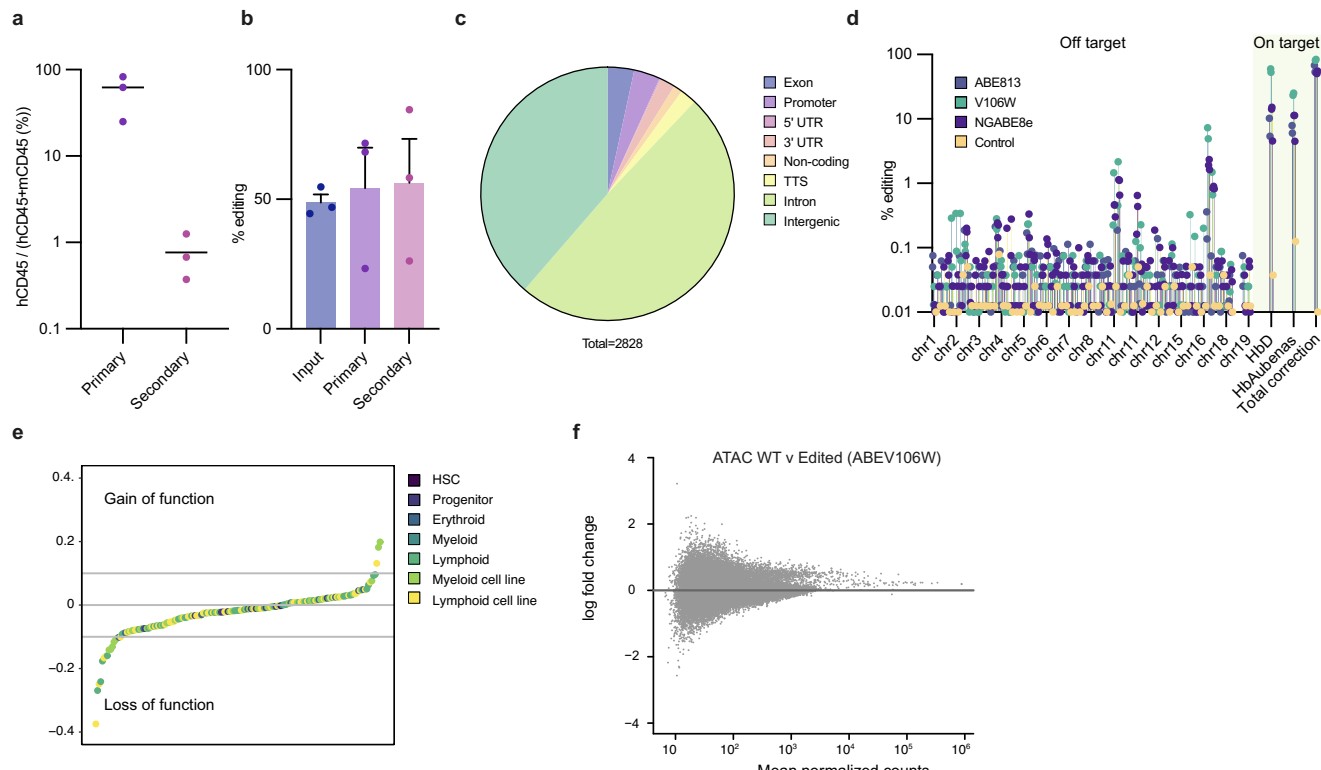

**Fig. 2 | Editing of long term haemopoietic stem cells and off target effects.**
**a** Engraftment of edited human cord blood HSCs in NSG mice in primary and secondary transplants ($n = 3$ biologically independent samples). **b** Editing efficiencies for the cells prior to transplantation and following the primary and secondary transplants ($n = 3$ biologically independent samples). These cells were edited with ABEmax, due to the lag time on these experiments and this has lower editing efficiencies than the ABE8 editors, which are also reflected in the in vitro (Supplementary Fig. 2d). **c** Location of all potential off-target effects identified by CIRCLE-seq combined with two in silico approaches (Cas-OFFinder and CRISPOR).
**d** Targeted sequencing through oligonucleotide capture at the top 250 sites identified, which confirmed likely low level off target editing at 70 sites.
**e** Predictions from DeepHaem of the effects of all possible off-target edits in the non-coding genome on chromatin accessibility. **f** MA plots based on ATAC-seq data comparing WT vs base edited cells, showing that there were no significant differentially accessible peaks detected (DESeq2, alpha 0.05). Source data are provided as a Source Data file.

three donors with β-thalassaemia mutations that do not disrupt the PAM, the WT sequence (GAG) was converted to the Hb Aubenas sequence (GGG) with a mean 74% efficiency. This is not predicted to result in any further deleterious effects as this allele has no or very low β-globin expression due to the pre-existing β-thalassaemia mutation. Indels at the on-target site were detected at minimal levels (mean 0.15% SD 0.06%).

CD34 + cells were differentiated into mature erythroid cells with no morphological or immunophenotypic perturbations in differentiation detected (Supplementary Fig. 3). Globin gene expression profiling by RT-qPCR showed that there was a significant increase in β-globin mRNA, with a mean 57.8% rise in expression (Fig. 1d). As the pathophysiology of HbE β-thalassaemia is driven by disordered protein production in erythroblasts, cation-exchange reverse-phase high performance liquid chromatography (CE-HPLC) was performed on control and edited patient cells. There was a significant reduction in the level of HbE when comparing control to edited cells (mean 59.1% SD 13.2% to 17.2% SD 5.4%), with a concurrent increase in HbA and Aubenas (7.0% SD 3.9% vs 57.6% SD 12.1%). This represented a 13.7-fold increase in normal or normal variant haemoglobins and a 3.1-fold decrease in HbE (Fig. 1f). As expected for patients with HbE β-thalassaemia, the level of HbF is increased and does not change in edited cells (Fig. 1e). In case editing produced variant haemoglobins not detected by HPLC, isoelectric focusing (IEF) was performed. There was no evidence of new haemoglobins being produced except for Hb Aubenas (Fig. 1g). Sequencing of poly A negative RNA from erythroid cells that have undergone correction of the codon results in reduced aberrant splicing of the transcript and persistent

off-target RNA editing was not detected in these cells (Supplementary Fig. 2g).

To show editing of LT-HSCs, serial murine xenograft transplantation assays were performed using the well characterised NSG mouse model[16]. Edited CD34 + human cells were detected following both primary and secondary transplants (Fig. 2a). No fall in mean editing efficiency was detected between primary and secondary transplants (Fig. 2b).

**Profiling of off-target effects**
Extensive profiling of off-target editing events was undertaken. The specificity of our sgRNA was profiled by combining *in-silico* methods with CIRCLE-seq (Fig. 2c)[17–19]. This combination of approaches identified 2829 potential / theoretical off-target sites genome-wide, of which 1399 had an adenine in the target window (Supplementary Data 1).

We went on to perform targeted oligonucleotide capture from the top 250 candidate sites identified, using targeted oligonucleotide capture followed by high throughput sequencing of patient samples that had undergone genome editing (Fig. 2d, Supplementary Data 2 and 3). This approach allowed highly sensitive profiling of the real off-target effects because each site was sequenced to an average read depth of 53,922. 3 base editors were used (ABE8.13, ABE8e-V106W, NG-ABE8e) with two technical replicates for each editor. Off-target editing was found at 70 sites but generally at very low levels (median 0.038%). The site with the highest editing was expected, at the highly homologous *HBD* gene, which has an identical sequence to the on-target site, and so had 52.9% deamination frequencies. *HBD* is expressed at a low level and forms HbA2 ($\alpha_2\delta_2$), which comprises 2-3% of total adult

haemoglobin and has no significant physiological function. All of the other off-target effects were located either in introns or intergenic regions. Off-target editing was detected at a maximum frequency of over 1% at two intergenic sites, each with 2 base pair changes in the editing window (hg19 chr18:76699400/76699402 & chr11 28480163/ 28480163). Both of these were located in intergenic regions (nearest gene 40.8 kb and 18.9 kb respectively) and neither of these sites were located in hypersensitive sites.

Little, if any, work has been done to look at the genomic consequences of these mutations. To address this, we combined analysis of functional genomics data with machine learning approaches, which we have previously used to successfully identify the effects of non-coding variants[20]. The deep learning model, DeepHaem is trained on over 600 datasets, including 49 different blood cell type datasets[21] and it is able to identify the effects of mutations in regulatory elements, particularly gain of function variants which would be missed by conventional approaches such as intersection with known hypersensitive sites (Supplementary Data 4 and 5).

At the two intergenic sites mentioned above (hg19 chr18:76699400/76699402 & chr11 28480163/28480163) with >1% OT-editing the machine learning model predicted that these changes would not alter chromatin activity.

Low level editing (0.07%) was seen in the promoter of *OGFOD2* but this was also not predicted by the machine learning model to cause damage and the gene is most highly expressed in sperm. In addition, variants in the gene are not associated with disease on ClinVar. Two lncRNAs (*ACTN1-AS1* exon 9 and *LINC01569* promoter) were potential off-targets but neither of these have any association with haematological disease[22]. The remaining intronic and intergenic sites at which editing had been identified were also predicted to be inert.

We went on to use the approach to analyse all 2829 sites where potential off-target editing might occur and only 17 sites were identified that were predicted to alter the chromatin state (Fig. 2e). None of these sites were near any genes that are recurrently mutated or dysregulated in haematological malignancy. In addition, we undertook ATAC-seq in both WT and edited HSPCs and found no differences in normalised peak counts (Fig. 2f, Supplementary Fig 4).

## Discussion

Here we show that Haemoglobin E, which causes 50% of all severe transfusion dependent thalassaemia worldwide, can be corrected to a non-pathogenic variant Hb Aubenas, using adenine base editors. A similar approach has been used to edit the mutation that causes Sickle Cell Disease, which can be deaminated to form Hb Makassar[23, 24]. This approach is advantageous to previous methods as it does not involve random integration of a highly active construct or involve generating potentially genotoxic double strand breaks. Using established and novel machine learning based methods, we have shown that base editing has a favourable off-target editing profile. We have not assessed the potential for sporadic off-target editing but this has previously been carefully characterised and found not to be a major problem albeit with lower activity earlier generation editors[25]. Base editing will therefore potentially prove to be the optimal way to cure the majority of patients with haemoglobinopathies.

## Methods
### Preparation of cells
Patients peripheral blood collection was performed at the Churchill Hospital, Oxford or Department of Paediatrics, University of Kelaniya, Sri Lanka using standard procedures, following written informed consent for collection for research. The study complies with all relevant ethical regulations and has approval from the Oxford South Central C Research Ethics Board; WIMM R&D committee (ref. 17/SC/ 0111) and the Sri Lanka College of Paediatricians. Human umbilical cord blood (UCB) was collected from the John Radcliffe Hospital, Oxford, UK or provided via the NHS Cord Blood Bank, London, and used with informed, written pre-consent and ethical approval (REC Ref. no. 15/SC/0027) from the South Central Oxford and Berkshire Ethical Committees and approval of the NHSBT R&D committee. The patients had a variety of genotypes (Supplementary Data 6).

### Isolation and CD34 + culture
Mononuclear cells (MNCs; density <1.077 g/ml) were isolated by density gradient centrifugation. Human CD34 + hematopoietic stem and progenitor cells (HSPC) were enriched by MACS using the CD34 direct microbead kits (Miltenyi Biotec GmbH). After isolation or thawing, HSPCs were placed in HSPC media comprised of StemSpan SFEM II (Stemcell technologies) supplemented with 100 ng/ml stem cell factor (SCF) (PeproTech), 100 ng/ml hrombopoietin (TPO) (PeproTech), 100 ng/ml fms-like tyrosine ligand 3 (FLT3L) (PeproTech) and 1 IU/ml penicillin/streptomycin (Gibco). HSPCs were seeded at a density of $0.25 \times 10^6$ cells/ml and cultured for 36-48 hours at 37 °C and 5% $CO_2$.

### Erythroid culture
CD34 + cells were differentiated down the erythroid lineage using a modification of a published differentiation protocol[26]. All phases used a prepared base media containing Iscove's modified Dulbecco's media (Bioscience UK), 200 μg/ml human holo-transferrin (HT) (R&D systems), 10 μg/ml recombinant human insulin (Sigma Aldrich), 3 IU/ml heparin sodium (Sigma Aldrich), 3% inactivated group AB Plasma (Department of Haematology, Oxford University Hospitals Trust), 3IU/ ml erythropoietin (Janssen-Cilag) and 2% foetal bovine serum.

Phase 1 (Day 0 to 7) – Freshly isolated or thawed HSPCs were seeded at a density of $2 \times 10^5$ cells/ml in base media supplemented with 10 ng/ml SCF and 1 ng/ml Interleukin-3 (Peprotech). Cells were counted every 48 hours from Day 3 onwards and media was added to dilute the cells to a concentration of $2 \times 10^5$ cells/ml.

Phase 2 (Day 7 to 10) – Cells were counted and the media was changed by centrifuging the cells for 5 minutes at 300 g. The cells were seeded at $2 \times 10^5$ cells/ml in base media supplemented with 10 ng/ml SCF. Cell density was maintained at $2 \times 10^5$ cells/ml.

### HUDEP-2 culture
HUDEP-2 cells kindly by Dr Kurita and Dr Nakamura from the RIKEN Tsukuba Branch were maintained at a concentration of $2.5 \times 10^5$ cells/ ml – $1.5 \times 10^6$ cells/ml in StemSpan SFEM (Stemcell technologies) supplemented with 2 mM glutamax (ThermoFisher), 1IU/ml penicillin/ streptomycin (Gibco), 50 ng/ml human stem cell factor (PeproTech), 3IU/ml erythropoietin (Janssen-Cilag), 840 nM dexamethasone (Hameln) and 2 μg/mL Doxycycline (Sigma Aldrich). Cells were counted every 48 hours, centrifuged at 300 g for 5 minutes and resuspended in fresh media at a concentration of $2.5 \times 10^5$ cells/ml.

### Production of ABE mRNA
Base editor plasmids were linearised using *AgeI* (NEB) at the 3' end of the editor sequence. Base editor mRNA transcription was performed using the mMESSAGE mMACHINE T7 Ultra Kit (Thermofisher) following the manufacturer's protocol. As the transcripts were longer than 5 kb 1 μl GTP was added to the reaction. Clean-up of the polyadenylated product was carried out with Megaclear™ Transcription clear kit (Invitrogen) according to the manufacturer's instructions and the mRNA was resuspended in 20 μl nuclease free water.

### Genome editing−CD34 + cells
After 48 h of culture in HSPC media, CD34 + cells were transfected using the P3 Primary Cell 4D-Nucleofector TM X kit (Lonza). An ABE mRNA-sgRNA solution was formed by mixing 50 pmol chemically modified synthetic targeting or scrambled control sgRNAs (Synthego) with 2.5 μg ABE8e mRNA at a molar ratio of 1:2.5 for

 

10 minutes at 23 °C. $1 \times 10^5$ HSPCs were resuspended in 20 µl P3 solution and were thoroughly mixed with the formed mRNA-sgRNA complex. The mixture was added to a 20 µl cuvette and electroporated using the DZ-100 program on the Lonza 4D Nucleofector. Immediately post-electroporation cells were placed in HSPC maintenance media for 24 h to recover and then transferred to erythroid differentiation culture media. Cells were harvested on Day 10 for downstream analysis.

## High-throughput sequencing of the *HBB* locus using NGS

Locus specific primers including adaptor sequences targeting exon 1 of *HBB* were used to quantify editing efficiencies using a modified 2-PCR version of the NEBnext Ultra II (NEB) library preparation protocol. Amplification was performed using the Herculase II (Agilent). See Supplementary Information for Primer sequences. In the first PCR, primer pairs NEB adaptors 5' to the locus-specific primer sequence were used for amplification, ensuring the products had the adaptor sequences added by the end of the PCR. 5 µl of this was used directly without clean-up for the second PCR. This indexing PCR added dual end indices taken from the NEBNext Multiplex Oligos for Illumina kit (NEB). Material was sequenced using the Illumina platform and sequences were extracted using the standard software (e.g. MiSeq Control Software v4.0).

## Globin gene expression quantification

RNA was extracted using the Rneasy mini kit (Qiagen) according to the manufacturer's instructions. Complementary DNA (cDNA) was produced from RNA using the SuperScript III First-Strand synthesis Supermix for qRT-PCR (ThermoFisher) according to the manufacturer's instructions. Predesigned and validated Taqman probes (Applied Biosystems) and Taqman Universal PCR mastermix (ThermoFisher) were used in all qPCR assays (TaqMan IDs: HBA2/HBA1-Hs00361191_g1, HBB-Hs00747223_g1, HBG-Hs00361131_g1 and RPL13A-Hs03043885_g1). All Taqman probes spanned exon junctions. Reactions were setup in 20 µl in technical triplicate and run on the Quantstudio 3 Real Time PCR System (Applied Biosystems). Gene expression was calculated using the delta delta CT method.

## RNA sequencing

RNA was extracted from edited CD34 cells differentiated in erythroid culture using the RNeasy Mini Kit (Qiagen), following the manufacturer's protocol. RNA quality was assessed by tape station, using RNA screentape (Agilent). Ribosomal RNA was depleted using the NEBNext rRNA Depletion Kit. Poly-A positive and negative fractions were separated using the NEBNext Poly(A) mRNA Magnetic Isolation Module. Poly-A positive and negative RNA-seq was then performed using the NEBNext Ultra II Directional RNA Library Prep Kit for Illumina (New England BioLabs). Sequencing was done using NovaSeq (Illumina) at 150 bp paired end.

## Protein quantification

For CE-HPLC $7 \times 10^6$ cells were used per replicate. The Bio-Rad variant haemoglobin testing system was used according to the manufacturer's instructions with the Variant II β-thalassaemia short program pack. IEF was performed as per the manufacturer's instructions (Resolve; PerkinElmer), running for 45 minutes at 300 V; 15 °C. $0.5 \times 10^6$ cells were used per sample.

## CIRCLE-seq

CIRCLE-seq was carried out comparing WT and edited cells using the previously described protocol[27] except that the NEBNext reagents (E7370L) were used to ligate the adaptor sequences. Off-target editing using the HbE targeting sgRNA was assessed in triplicate, using the HUDEP-2 HbE line and DNA from two patients with HbE β-thalassaemia.

## Oligonucleotide hybridization, capture and sequencing

Briefly, sequencing adaptors were added using the NEB Ultra II kit and the libraries were amplified by PCR (Herculase II, Agilent) to add indexing sequences. In total 10 µg of libraries was pooled for each hybridization reaction. The Roche SeqCap hybridization reagents and protocol were followed protocol. The hybridization reactions and bead washes were scaled such that for each 1–2 µg of library used, 5 µg human COT DNA and 1000 pM Nimblegen HE index-specific blocking oligonucleotides were used. This mixture was denatured by heating to 95 °C for 10 min before being hybridized for 72 h with 120-bp biotinylated oligonucleotides at a concentration of 130 fmol per sample. The samples were captured with streptavidin beads (Thermo Fisher, M270), washed and amplified as per protocol. A second round of oligonucleotide capture was performed with the same oligonucleotides and reagents with only a 24-h hybridization reaction. The material was sequenced on the Illumina NovaSeq with 300-bp reads (150-bp paired-end reads).

## Mouse xenograft assays

Experiments were performed under the project license P8869535A approved by the UK Home Office under the Animal (Scientific Procedures) Act 1986 and in accordance with the principles of 3Rs (replacement, reduction and refinement) in animal research and mice were euthanised by a schedule 1 approved method (dislocation of the neck under terminal anaesthesia). 100,000 cord blood CD34 + cells from three different biological donors were electroporated and kept in culture medium for 24 hours. Cells were then washed and resuspended in PBS + 1% FBS and injected via the tail vein into sub-lethally irradiated female NSG (NOD.Cg-Prkdcscidll2rgtmlWjl/SzJ; Jackson laboratories) mice. Mice were monitored daily. 16 weeks post transplantation, mice were euthanised and human grafts were analysed by flow cytometry (detailed in Supplementary Information).

## Assay for transposase-accessible chromatin (ATAC)−sequencing

The protocol was adapted for small cell numbers from Buenrostro et al[28]. Cells were harvested into 50 µL of cold lysis buffer (10 mM Tris-HCl, pH 7.4, 10 mM NaCl, 3 mM $MgCl_2$, 0.1% IGEPAL CA-630) and spin down at 4 °C for 10 minutes at 500 g. Nuclei were then resuspended in 50 µl transposition reaction mix (25 µl TD buffer, 2.5 µl Tn5 and 22.5 µl water). Post incubation for 30 min at 37 °C, Transposed fragments were purified using a Qiagen MinElute Kit into 23 µl elution buffer. Purified fragments were then amplified by PCR as previously described.

## Statistics and reproducibility

Editing experiments were performed on 6 biological replicates to demonstrate that consistently high editing efficiencies are possible. Randomisation and blinding was not undertaken for these experiments. Staff in the animal facility were blinded to the experimental conditions.

No statistical method was used to predetermine sample size. No data were excluded from analyses. Experiments were not randomised and the investigators were not blinded to allocation during experiments and outcome assessment.

## Data analysis

CasOFFfinder v2.4 (http://www.rgenome.net/cas-offinder/)[17] and CRISPOR[18] were used to define potential gRNA related off target sites in silico. CIRCLE-seq sequencing data was analysed using the circleseq Python package (https://github.com/tsailabSJ/circleseq). Capture oligonucleotide design: 120-bp oligonucleotides were designed to capture the off-target sequence using CapSequm (https://github.com/jbkerry/capsequm). Off-target capture data were analysed using Trim Galore (Babraham Institiute, v0.3.1) and FLASH (v1.2.11)[29] and mapped

to the genome (hg19) using Bowtie 2 (v2.3.5)[30]. Samtools mpileup was used to call variant bases and a custom script was used to identify likely off-target editing (https://github.com/jojdavies/Base_editing_off-targets).

RNA-seq data was aligned using STAR (2.7.3a) to hg19[31]. Aberrant splicing of Poly-A negative RNA was detected using PySam (https://github.com/pysam-developers/pysam) find_introns method, considering position 5,248,159 of chromosome 11 (hg19) as the canonical splice site for exon 1 of the *HBB* gene. An RNA variant calling pipeline was established using GATK best practices[32]. In keeping with the Broad Institutes recommendation for this tool, these data were realigned to hg38 using STAR two-pass alignment, followed by PCR duplicate removal and base score recalibration. GATK Haplotype Caller (4.0.11.0) was then used to call variants. ATAC-seq data was aligned using Bowtie2 and Peak Called with MACS2. DEseq2 was run using default parameters to assess whether any peaks were significantly different between edited and control samples. The false discovery rate/alpha-value was set at 0.05.

The DeepHaem Machine learning model was adapted for determining the effects of non-coding off-targets[20]. The model was trained on chromatin accessibility and ChIP-seq datasets generated from haematological cell types (see Supplementary Data for details)[33]. All potential off-target sites identified by CIRCLE-seq and in silico methods were analysed. Initially sites were removed which did not contain an adenine in the editing window. All remaining potential base editing off-target effects within the targeting windows were then analysed using the model, which only requires the DNA sequence as an input. A P(accessible) score of 0.2 denotes a site likely to be in an accessible chromatin site in-vivo. All sites with a P(accessible) > 0.2 were selected. For every site a damage score was calculated (P(accessible)$_{control}$ – P(accessible)$_{edited}$). Damage scores greater than 0.1 are likely to be significant, and so all sites with a P(accessible)$_{control}$ > 0.2 and damage score of >0.1 were used for further assessment. Data were visualised using Prism (9.5.0) and Rstudio (1.2.5033).

## Reporting summary

Further information on research design is available in the Nature Portfolio Reporting Summary linked to this article.

## Data availability

Sequencing data has been submitted to the NCBI Gene Expression Omnibus (GSE206098). All data generated or analysed during this study are included in this published article (and its supplementary information files). Source data are provided with this paper.

## Code availability

All custom scripts are available on GitHub (https://github.com/jojdavies/Base_editing_off-targets). The code for the machine learning models to predict the effects of off-target mutations in the non-coding genome is also available on GitHub (https://github.com/rschwess/deepHaem).

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

## Acknowledgements

M.B. is supported by an Medical Reaserch Council (MRC) Clinical Research Training Fellowship (MR/P019633/1) and J.D. and P.H. were funded by an MRC Clinician Scientist Award (MR/R008108) to J.D and the MRC Molecular Haematology Unit (MC_UU_00029/04). This work was also supported by an MRC Discovery Award led by D.H. (MC_PC_15069) and an MRC project grant (MR/T030410/1) supported W.Z. J.D. and L.H. are supported by the Oxford National Institute of Health Research Biomedical Research Centre (NIHR203311). J.D. is also supported by the National Institute of Health Research Blood and Transplant Research Unit in Precision Cellular Therapeutics (NIHR203339). A.E. and F.I. are supported by Wellcome Fellowships (102176/B/13/Z and 211122/Z/18 respectively). J.H. developed machine learning approaches with support from the National Institutes of Health (USA) grant number R24DK106766 and he is supported by the MRC Molecular Haematology Unit (MC_UU_00016/14). J.D. and J.H. are also supported by Wellcome (225220/Z/22/Z). Dr Kurita and Dr Nakamura from the RIKEN Tsukuba Branch kindly provided HUDEP-2 cells. We would like to thank the staff at the WIMM who were essential for the completion of the project, including Philip Hublitz, Caroline Scott, Kevin Clark, Tim Rostron, John Frankland, Sue Butler, Jackie Sloane-Stanley, Sue Harper, Tim Quantick, Carol Eaton, Noelle Obers, Oliver Burns and Stella Keeble. Finally we would like to thank the patients who donated their time and samples towards this work.

## Author contributions

J.D. and M.B. conceived the project, designed, performed, analysed experiments, performed the majority of bioinformatic analyses and wrote the first draft of the manuscript. A.E., P.H., S.R., W.Z., C.A.F., N.D., F.I., A.R., L.D.H. analysed data and performed experiments. N.R., P.T., N.Y. and S.M. provided patient samples. J.H. and R.S. developed the deep learning models. D.H. provided funding and assisted with experimental design. All authors contributed to writing the manuscript.

## Competing interests

J.D., M.B. and D.H. have filed a patent application on this work,[24] which has been licensed to BEAM therapeutics. J.D. and M.B. receive revenue from this licence and hold personal shares in BEAM therapeutics. J.D. and J.H. are co-founders of Nucleome Therapeutics Ltd. and provide consultancy to the company. The remaining authors declare no competing interests.
