## [Peer Review File · Nature Communications]

REVIEWER COMMENTS

Reviewer #1 (Remarks to the Author):

In this report, Badat et al. demonstrate base editing correction of the hemoglobin E mutation using the ABE8e base editing system. This yields a combination of minority normal and majority benign variant (hemoglobin Auenas) beta-globin alleles. The correction is efficient in primary CD34+ HSPCs from patients and rescues globin gene expression and hemoglobin production in erythroid progeny in vitro. Edited cells may engraft immunodeficient mice in primary and secondary recipients, although relatively scant in vivo data is shown and editing efficiency appears lower as compared to in vitro studies. gRNA-dependent off-target effects are extensively evaluated, and these are mainly infrequent, with none predicted to be functional. A machine-learning approach is applied to evaluate for potentially functional off-target effects, which appears to be a novel method to prioritize candidate off-targets, and might be more widely adapted by other investigators for genome editing off-target analyses. Overall this paper describes a promising base editing approach that can be added to the growing list of genetic maneuvers to rectify beta-hemoglobinopathies and could be clinically impactful. In addition, improved functional prioritization of candidate off-target effects would be a useful contribution to the gene editing field.

Major comments:

1. A concern is in vivo experiments seem to show less efficient editing as compared to in vitro experiments. For the engraftment experiments in Fig 2a/b, how many donors? How many mice per donor? (Also how many donors for the in vitro experiments of Fig 1?) The variation in editing frequencies in engrafted cells appears quite large. Why is input editing frequency in Fig 2a ~50% but ~85% in Fig 1b, raising questions about reproducibility of high editing efficiencies seen in Fig 1b?
2. It is unclear for the functional experiments in Fig 1 d-g how many donors and what are their beta-thalassemia genotypes? At a minimum experiments should be shown from at least one donor with a beta-globin genotype that allows non-target strand editing and from at least one donor with a beta-globin genotype that prevents non-target strand editing. It seems possible that the level and type of non-target strand editing could modify hemoglobin expression, by producing clones with homozygous hemoglobin Auenas or with altered splicing patterns.
3. The "DeepHaem" method could be more widely useful for the gene editing field to prioritize potentially functional noncoding off-target effects for further validation. Based on the description, it was unclear to me what are the required inputs to perform this analysis (is it just a list of off-target sites? Also relevant cell types? Predicted edits?). Also I was not sure that based on the brief description provided if this novel method could be easily reproduced and extended to other gRNAs and off-target lists by other investigators. I suggest expanding the description of the method and ensuring it could be reproduced and utilized by others, perhaps by providing a manual/more extensive description, test data, and expected outputs.

Additional comments:

4. The expected target population for the editing would be patients with compound heterozygous hemoglobin E and beta-thalassemia, since homozygous hemoglobin E is not a serious disease. Given that most of the edits convert the hemoglobin E allele to a variant hemoglobin Aubenas allele, the expected final hematology if successful would be compound heterozygous hemoglobin Aubenas and beta-thalassemia. What is known about this condition? Is it any more severe than heterozygous beta-thalassemia carrier status? How about homozygous Hb Aubenas? If there is no such clinical data available, this should be clearly stated as a limitation and possible concern for the approach.

5. Related question, what is being plotted in ED Fig 1a? Is this Hb concentration, MCV, MCH? Are the numbers from bE/bA and bAubenas/bA reflecting the mean or a range? From how many subjects? Age/sex? Please add more details to figure and legend.

6. The authors claim hemoglobin Aubenas prevents activation of a cryptic splice donor unlike hemoglobin E. Could they show their analysis that supports this prediction?

7. For Fig 1c, add to figure legend the genotype of the thalassemic allele for the donors tested. Presumably this result is from the IVS1-5 mutation that disrupts the PAM sequence and prevents non-target strand editing.

8. Did the authors evaluate gRNA-independent RNA and DNA editing? At the very least, this should be discussed as a potential risk of base editing.

9. A panel similar to Fig 1c showing the distribution of alleles should be provided for input and primary and secondary engrafted samples. This could be in a table or on a log scale or with split axis to make allele frequencies visible across the frequency range.

10. Does the editing procedure impair engraftment? No unedited control is shown. One theoretical concern could be mRNA delivery to HSCs could induce an interferon-response which could limit engraftment.

11. Why was a mouse model used that doesn't support human erythroid engraftment? It would have been informative to see the hemoglobin expression in engrafting erythroid cells. At least this could be discussed as a limitation.

12. What was the on-target editing level in the target-capture off-target experiment? Perhaps this could be included in the supplemental table of off-target editing with a row for the on-target editing.

13. What is the sensitivity of the target-capture assay?

14. What level of indels were seen in the various in vitro and in vivo experiments across donors and replicates? (Data might need to be plotted on a log scale or with a split axis to show this alongside high-level on-target base edits.) Typically a low-level of indels are observed with base editing, including ABE8e (e.g. Richter et al. NBT 2020).

Reviewer #2 (Remarks to the Author):

This manuscript by Badat et al presents a strategy for correcting haemoglobin E/ β -thalassaemia using next-generation Adenine base editors (ABC)s. They show that they can employ these ABCs to repair the HbE mutation due to a conveniently located TGG PAM site within range with 85% efficiency. The authors rigorously evaluated potential off-target methods using wet-lab techniques like CIRCLE-SEQ followed by targeted amplification revealing that they occurred rarely. The authors further explored these off-target sites utilizing deep learning methods revealing that they likely will not have a functional effect in these cells. The authors do a wonderful job explaining and presenting their results. Below are some minor revisions to improve upon before publication.

1. The figure labels, particularly figure 1, have some compression artifacts. Ensure that they are accurate when typesetting.
2. I saw no data that measured the delivery efficiency of this protocol. Was it that the ABC was delivered to 100% of cells and only edited 85%, or was it that the ABC only reached 90% of cells and then edited 95% of the cells that it reached? This will help to understand whether the need for improvement lies in improving the ABC or improving the delivery methodology.
3. While I like the use of CIRCLE-seq as an in-vitro genome wide screen, I am skeptical of its application in this context. The technique requires the cutting of the circle while A-base editing only requires the binding of the Cas9, a thermodynamically easier and more common action than cutting (PMC4732943). In a brief search, I was not able to find any research exploring whether CIRCLE-seq appropriately captures the full scope of off-target editing by A-base editors. The authors should provide references using CIRCLE-seq for ABC off-target evaluation or otherwise comment on it.

Response	to	reviewers	comments
Reviewer	#1	(Remarks to the	Author):

In this report, Badat et al. demonstrate base editing correction of the hemoglobin E mutation using the ABE8e base editing system. This yields a combination of minority normal and majority benign variant (hemoglobin Aubenias) beta-globin alleles. The correction is efficient in primary CD34+ HSPCs from patients and rescues globin gene expression and hemoglobin production in erythroid progeny *in vitro*. Edited cells may engraft immunodeficient mice in primary and secondary recipients, although relatively scant *in vivo* data is shown and editing efficiency appears lower as compared to *in vitro* studies. gRNA-dependent off-target effects are extensively evaluated, and these are mainly infrequent, with none predicted to be functional. A machine-learning approach is applied to evaluate for potentially functional off-target effects, which appears to be a novel method to prioritize candidate off-targets, and might be more widely adapted by other investigators for genome editing off-target analyses. Overall this paper describes a promising base editing approach that can be added to the growing list of genetic maneuvers to rectify beta-hemoglobinopathies and could be clinically impactful. In addition, improved functional prioritization of candidate off-target effects would be a useful contribution to the gene editing field. We thank the reviewer for their careful review of our work and their positive comments on the manuscript.

Major comments:

1. A concern is *in vivo* experiments seem to show less efficient editing as compared to *in vitro* experiments. For the engraftment experiments in Fig 2a/b, how many donors? How many mice per donor? (Also how many donors for the *in vitro* experiments of Fig 1?) The variation in editing frequencies in engrafted cells appears quite large. Why is input editing frequency in Fig 2a ~50% but ~85% in Fig 1b, raising questions about reproducibility of high editing efficiencies seen in Fig 1b?

We thank the reviewer for picking up on this and apologise that this discrepancy wasn't explained clearly in the manuscript (we have now clarified this). Due to the prolonged lag time involved with murine xenotransplantation experiments, the cells used for these experiments were edited with one of the previous generation of ABE editors, ABEmax, which does not have as good editing efficiency as the 8th generation adenine base editors used in figure 1. The work describing the latter editors was published after our murine experiments had commenced. Of note the editing efficiencies with ABEmax are very similar in the *in vitro* (Extended Data Fig. 2d) and *in vivo* experiments. We have repeated the experiments in Fig 1 with a further 3 donors and have reconfirmed the high efficiencies.

2. It is unclear for the functional experiments in Fig 1 d-g how many donors and what are their beta-thalassemia genotypes? At a minimum experiments should be shown from at least one donor with a beta-globin genotype that allows non-target strand editing and from at least one donor with a beta-globin genotype that prevents non-target strand editing. It seems possible that the level and type of non-target strand editing could modify hemoglobin expression, by producing clones with homozygous hemoglobin Aubenias or with altered splicing patterns.

For the initial *in vitro* experiments 2 donors with HbE-IVS I-V and one donor with HbE-CD16 were analysed. In order to improve this aspect of the work we have now undertaken experiments from an additional 3 patients so that we now have analysed a total of 6 different patients half of whom are non-IVS I-V genotypes, which do not disrupt the PAM sequence. The details of the genotypes are now included in the supplemental material.

We have also undertaken additional analysis of the HBD gene, which allows the activity of the gRNA to be determined for the WT sequence across all of the patients including the IVS I-V genotype. This shows that there are similar levels of editing of the beta-thalassaemic and HbE alleles when the PAM is not disrupted. We do not think that this will be a significant problem because the beta thalassaemic allele, is not expressed and the HBD gene is only expressed to a very limited extent.

3. The "DeepHaem" method could be more widely useful for the gene editing field to prioritize potentially functional noncoding off-target effects for further validation. Based on the description, it was unclear to me what are the required inputs to perform this analysis (is it just a list of off-target sites? Also relevant cell types? Predicted edits?). Also I was not sure that based on the brief description provided if this novel method could be easily reproduced and extended to other gRNAs and off-target lists by other investigators. I suggest expanding the description of the method and ensuring it could be reproduced and utilized by others, perhaps by providing a manual/more extensive description, test data, and expected outputs.

DeepHaem only requires a list of candidate off-target sites as the input. The curation of this list would be left to the end-user; in our case, we used two common *in-silico* methods (Cas-OFFinder and CRISPOR) as well as CIRCLE-seq to identify as many potential sites of off-target editing as possible. These were filtered for sites where there was an Adenine in the target window and we compared the potential edited sequence with the WT. The deep learning model initially included 694 datasets but we restricted the classifiers to 49 blood / bone marrow derived cell types, which included leukaemia cell lines. A list of cell types the model has been trained on, and can therefore provide predictions for, has now been provided in Supplemental Table 2. The code is available in GitHub (<https://github.com/rschwess/deepHaem>).

We thank the reviewer for these comments and have expanded the description of the code and required inputs.

Additional comments:

4. The expected target population for the editing would be patients with compound heterozygous hemoglobin E and beta-thalassemia, since homozygous hemoglobin E is not a serious disease. Given that most of the edits

convert the hemoglobin E allele to a variant hemoglobin Aubenas allele, the expected final hematology if successful would be compound heterozygous hemoglobin Aubenas and beta-thalassemia. What is known about this condition? Is it any more severe than heterozygous beta-thalassemia carrier status? How about homozygous Hb Aubenas? If there is no such clinical data available, this should be clearly stated as a limitation and possible concern for the approach.

At present Hb Aubenas has only been described in the heterozygous state in one family in France. The original description was published in 1996 and we contacted Henri Wajcman to obtain further clinical details. Unfortunately, none of the family members are under clinical follow up and our attempts to contact them were unsuccessful. The original paper describes a 17 year old girl in detail. One of her siblings was also found to have the mutant haemoglobin and although her parents both refused Hb analysis it is almost certain that one parent would also have been a carrier. It is likely therefore that there are at least 3 cases of asymptomatic heterozygous carriage of Hb Aubenas.

There are reasons to suggest that a β Aubenas/ β Thal0 would not have a significant haematological phenotype. As shown in supplemental figure 1, the amino acid change is not predicted to cause major structural changes to the molecule. We have now done RNA-seq analysis and shown that β Aubenas is not transcribed at a different rate compared to normal, and in the heterozygous state the haematological indices are completely normal (in contrast to individuals with β -thalassaemia trait) so we would predict homozygotes to be normal. Similarly, as the β Aubenas chain is not significantly structurally different from β A, and is transcribed normally, β Aubenas/ β thalassaemic allele would be predicted to have the same haematological phenotype as β A/ β thalassaemia i.e. that of an asymptomatic carrier. However as the reviewers correctly note whilst these are reasonable hypotheses clinical data for these genotypes is not available and so a reference to this has been made in the manuscript.

5. Related question, what is being plotted in ED Fig 1a? Is this Hb concentration, MCV, MCH? Are the numbers from bE/bA and bAubenas/bA reflecting the mean or a range? From how many subjects? Age/sex? Please add more details to figure and legend.

We apologise - the top of this figure was removed in the formatting of the document. The reviewer is correct the plot shows Haemoglobin, MCV and MCH. These data were taken from a previously published report describing the respective genotypes. The figure legend has been modified to clarify this, along with the abbreviations used.

6. The authors claim hemoglobin Aubenas prevents activation of a cryptic splice donor unlike hemoglobin E. Could they show their analysis that supports this prediction?

In the revised manuscript we have provided more detail and undertaken RNA-seq to investigate the effects of genome editing on splicing HbE. First, we have included details of the in-silico predictions from Splice-AI, which predicts that HbE creates a cryptic splice site and that HbAubenas would reduce the potential for aberrant splicing compared to WT (Extended Data Fig. 1).

In response to the reviewer's comment we went on to perform RNA-seq on both Poly A selected RNA and on ribosomal depleted Poly A- RNA from edited cells that were differentiated to the erythroblast stage. This shows that base editing of the HbE allele significantly increases the proportion of reads that are spliced correctly.

7. For Fig 1c, add to figure legend the genotype of the thalassaemic allele for the donors tested. Presumably this result is from the IVS1-5 mutation that disrupts the PAM sequence and prevents non-target strand editing.

We thank the reviewer for this comment and we have now added data from additional genotypes, separating IVS1-5 from non-PAM disrupting genotypes. However, we suspect that this will not be of significant practical concern because the thalassaemic allele is not expressed significantly in the majority of these patients.

8. Did the authors evaluate gRNA-independent RNA and DNA editing? At the very least, this should be discussed as a potential risk of base editing.

We have undertaken further editing experiments and have evaluated gRNA independent RNA editing. We have not assessed gRNA independent DNA editing due to the difficulties in assessing this with whole genome sequencing but have added this as a caveat to the manuscript.

9. A panel similar to Fig 1c showing the distribution of alleles should be provided for input and primary and secondary engrafted samples. This could be in a table or on a log scale or with split axis to make allele frequencies visible across the frequency range.

The cells used in this assay were WT and so the editing shows % of Aubenas generated.

10. Does the editing procedure impair engraftment? No unedited control is shown. One theoretical concern could be mRNA delivery to HSCs could induce an interferon-response which could limit engraftment.

We agree with the reviewer and we believe that the editing procedure impairs engraftment to some extent and that the interferon response in HSCs is activated by mRNA delivery. We have demonstrated that it is possible to get editing of long-term HSCs with this approach but think that further work is required to minimise the toxicity to HSCs.

11. Why was a mouse model used that doesn't support human erythroid engraftment? It would have been informative to see the hemoglobin expression in engrafting erythroid cells. At least this could be discussed as a limitation.

We used the NSG mouse model because we simply wanted to demonstrate engraftment of LT repopulating HSCs and these models provide the classical accepted approach for establishing this.

12. What was the on-target editing level in the target-capture off-target experiment? Perhaps this could be included in the supplemental table of off-target editing with a row for the on-target editing.

The on-target editing was 63% on average and 80% for ABE8e106W. This was included in figure 2d and the associated data table but we have tried to make this more clear in the figure.

13. What is the sensitivity of the target-capture assay?

The target capture assay has a read depth of 53,922 across all replicates. We reported changes in the editing window when more than one read was present with the variant. It is likely to be sensitive to off-target edits at 0.01%.

14. What level of indels were seen in the various in vitro and in vivo experiments across donors and replicates? (Data might need to be plotted on a log scale or with a split axis to show this alongside high-level on-target base edits.) Typically a low-level of indels are observed with base editing, including ABE8e (e.g. Richter et al. NBT 2020).

These data are now included in Figure 1c. We have reassessed this in the light of the reviewer's comments and on reappraisal of the data we have only found indels at 0.15%.

Reviewer #2 (Remarks to the Author):

This manuscript by Badat et al presents a strategy for correcting haemoglobin E/ β -thalassaemia using next-generation Adenine base editors (ABC)s. They show that they can employ these ABCs to repair the HbE mutation due to a conveniently located TGG PAM site within range with 85% efficiency. The authors rigorously evaluated potential off-target methods using wet-lab techniques like CIRCLE-SEQ followed by targeted amplification revealing that they occurred rarely. The authors further explored these off-target sites utilizing deep learning methods revealing that they likely will not have a functional effect in these cells. The authors do a wonderful job explaining and presenting their results. Below are some minor revisions to improve upon before publication.

We are very grateful to the reviewer for taking the time to review our manuscript and their very positive comments.

1. The figure labels, particularly figure 1, have some compression artifacts. Ensure that they are accurate when typesetting.

We apologise for this and have uploaded the figures in a different format.

2. I saw no data that measured the delivery efficiency of this protocol. Was it that the ABC was delivered to 100% of cells and only edited 85%, or was it that the ABC only reached 90% of cells and then edited 95% of the cells that it reached? This will help to understand whether the need for improvement lies in improving the ABC or improving the delivery methodology.

In our most recent experiments which we performed in response to the comments from reviewer 1, we have editing efficiencies consistently over 95% so we believe that the delivery of the editing machinery is highly efficient.

3. While I like the use of CIRCLE-seq as an in-vitro genome wide screen, I am skeptical of its application in this context. The technique requires the cutting of the circle while A-base editing only requires the binding of the Cas9, a thermodynamically easier and more common action than cutting (PMC4732943). In a brief search, I was not able to find any research exploring whether CIRCLE-seq appropriately captures the full scope of off-target editing by A-base editors. The authors should provide references using CIRCLE-seq for ABC off-target evaluation or otherwise comment on it.

CIRCLE-seq has been used in combination with in silico approaches to screen off target base editing in a recent paper from David Liu's group (Newby et al 2021 <https://www.nature.com/articles/s41586-021-03609-w>). We agree that CIRCLE-seq may under estimate the potential sites, which is why we combined this with three different in silico approaches.

REVIEWERS' COMMENTS

Reviewer #1 (Remarks to the Author):

The authors have addressed my concerns.

Line 114 should be ED Fig. 2.

Reviewer #2 (Remarks to the Author):

All previous concerns have been addressed, manuscript is acceptable for publication.

REVIEWERS' COMMENTS

Reviewer #1 (Remarks to the Author):

The authors have addressed my concerns.

Line 114 should be ED Fig. 2.

Response: Thank you for picking this up, it has been amended.

Reviewer #2 (Remarks to the Author):

All previous concerns have been addressed, manuscript is acceptable for publication.